# Q-Match: Self-supervised Learning for Tabular Data by Matching Distributions induced by a Queue

## Abstract

In semi-supervised learning, student-teacher distribution matching has been successful in improving performance of models using unlabeled data in conjunction with few labeled samples. In this paper, we aim to replicate that success in the self-supervised setup where we do not have access to any labeled data during pre-training. We show it is possible to induce the student-teacher distributions *without* any knowledge of downstream classes by using a queue of embeddings of samples from the unlabeled dataset. We show that Q-Match outperforms previous self-supervised learning techniques on tabular datasets when measuring downstream classification performance. Furthermore, we show that our method is sample efficient, both in terms of labels required for both downstream task training and amount of unlabeled data required for pre-training.

## 1 Introduction

Tabular data is the most common form of data for problems in industry. While many robust techniques exist to solve real-world machine learning problems on tabular data, most of these techniques require access to labels. Leveraging unlabeled data to learn good representations remains a key open problem in the tabular domain. In this work, we propose a flexible and powerful framework using deep learning that helps us use unlabeled data in the tabular domain.

Deep learning has been successful in processing data in many different domains like images, audio, and text. Learning non-linear features using deep architectures has been shown to be a key feature in improving performance across a wide variety of problems like image recognition (Krizhevsky et al., 2017), speech recognition (Deng et al., 2013), and machine translation (Singh et al., 2017). Until recently, achieving state-of-the-art performance would not have been possible without large, manually annotated datasets. However, a class of learning algorithms called self-supervised learning (Wu et al., 2018; Devlin et al., 2018) has shown that highly performant features can be learned without large-labeled datasets as well. In this work, we propose a new self-supervised learning algorithm and study its effectiveness in the tabular data domain.

In self-supervised learning, a task known as the *pretext* task is first solved on a dataset that is typically large and unlabeled. The aim of self-supervised learning is to learn an encoding function $f$ parameterized by $\theta$ that captures variances and invariances in the dataset without the need for human-annotated labels. This initial training is usually referred to as the pre-training stage. The parameters learned from the pretext task during pre-training are then used to solve a new objective called the *downstream* task. Concretely, $f$ is a function with parameters $\theta$ that maps a sample $x$ from the input dimensionality $d'$ to an embedding size $d$ that is $f : (X, \theta) \mapsto R^d$ where $x$ is a single data-point in the input space $X$. After the pretext algorithm updates the model parameters $\theta$ during the pre-training stage, the downstream data (typically a manually annotated dataset) is either used to fine-tune $\theta$ or learn the parameters of a linear classifier on the output of $f$.

Recently proposed self-supervised approaches for tabular data (Darabi et al., 2021; Yoon et al., 2020; Arik & Pfister, 2021; Lee et al., 2020) have shown encouraging results. In this work, we propose a novel method for self-supervised learning called Q-Match which is closely related to the semi-supervised learning framework called FixMatch (Sohn et al., 2020). For labeled data, FixMatch uses the standard supervised learning loss. For unlabeled data, FixMatch proposes to match the student and teacher distributions over the set of classes used in the downstream task. In a

self-supervised setup, however, access to the relevant classes or any labeled data during pre-training is limited. For this reason, Q-Match uses a queue of embeddings instead of known classes to induce the teacher and the student distributions. The queue is implemented similar to MoCo He et al. (2020) and NNCLR (Dwibedi et al., 2021) by updating a list of embeddings with newer embeddings and discarding older ones as training proceeds. We use individual samples (via their embeddings) to generate the student and teacher distributions required for training. We find Q-Match leads to improvements not only in the final performance of the downstream model, but also reduces the number of labeled samples required for the downstream task.

## 2  PROPOSED APPROACH

**Motivation.** Our self-supervised approach is based on the success of the semi-supervised learning algorithm FixMatch (Sohn et al., 2020). In their method, the authors show it is possible to leverage a large unlabeled dataset and a few labeled samples to improve the performance of the model on a downstream task. They do so by matching the class distributions of the student and teacher *views* produced by augmenting the input in two different ways. We hypothesize that it might be possible to adapt their framework to the self-supervised setup by removing the dependency on the known classes during training. To do so, we keep a queue of past embeddings that can serve as a proxy for *classes*. We use this queue to produce the target and student distributions used to train a model. The training then proceeds by performing continuous knowledge distillation from the teacher to the student model such that the student ultimately learns to predict the distribution induced by the teacher.

**Method.** In Figure 1, we outline our method for the pretext training approach used in Q-Match. We corrupt the input $x_i$ two times independently using the method proposed in VIME (Yoon et al., 2020) to produce the student view $x_{i,s}$ and the teacher view $x_{i,t}$. We pass $x_{i,s}$ through the student model to produce the student embedding $z_{i,s}$. Similarly, we pass $x_{i,t}$ through the teacher model to produce the teacher embedding $z_{i,t}$. We want this pair of embeddings to induce similar probability distributions over a representative set of samples of the dataset. In other words, we want our encoder to maintain similar relationships between different data samples in spite of the corruption introduced while generating the views. As the dataset can be quite large, we maintain a fixed size queue $Q$ of past embeddings like MoCo (He et al., 2020) and NNCLR (Dwibedi et al., 2021). A stop-gradient is applied to the teacher embeddings, so only the student weights are updated at each iteration. We multiply the student embeddings and the teacher embeddings with the embeddings in the queue to produce the student and teacher logits. After taking softmax of these logits, we produce the student distribution $p_{i,s}$ and teacher distribution $p_{i,t}$ respectively. The teacher distribution $p_{i,t}$ is the target distribution which the student distribution $p_{i,s}$ should match. We define the distribution matching loss as follows:

$$\mathcal{L}_i^{\text{QM}} = H(p_{i,t}, p_{i,s}) = H\left(\text{softmax}\left(\frac{z_{i,t} \cdot Q}{\tau_t}\right), \text{softmax}\left(\frac{z_{i,s} \cdot Q}{\tau_s}\right)\right)$$

where $\tau_t$ is a scalar temperature value that controls the sharpness of the distribution produced by the teacher, $\tau_s$ is a scalar temperature value that controls the sharpness of the student distribution, $Q \in \mathbb{R}^{d \times m}$ is the queue of $m$ many previous teacher embeddings, and $H(p, q)$ is the cross-entropy loss between two distributions $p$ and $q$. We normalize the views $z_{i,s}$ and $z_{i,t}$ using L2 normalization. The queue size is constant and is refreshed at every training iteration with the previous batch of teacher embeddings while the oldest embeddings are removed from the queue. The parameters of the teacher model are updated using the Exponential Moving Average (EMA) of the student model parameters. Empirically, we observe that corrupting the teacher with a smaller probability leads to better performance (discussed later in Section 3.3).

**Model Architecture.** In all experiments, we use an MLP as our encoding function $f$. We follow the same architecture from i-Mix (Lee et al., 2020) which includes five fully connected layers (2048-2048-4096-4096-8192). The final layer uses a 4-set max-out activation (Goodfellow et al., 2013). All layers except the output layer have batch normalization followed by a ReLu. We use a linear projector of 128 dimensions on top of the encoder in all our experiments. Similar to i-Mix, we added a 2-layer (512-128) MLP projection head, but we noticed that it performed similar to the model without the projection head for  Q-Match training.

**Data Preprocessing.** For data preprocessing, we compute the normalization statistics by using a batch normalization layer without the learnable parameters of scale and bias just after the input layer. During evaluation, the accumulated exponential moving average statistics are applied to the

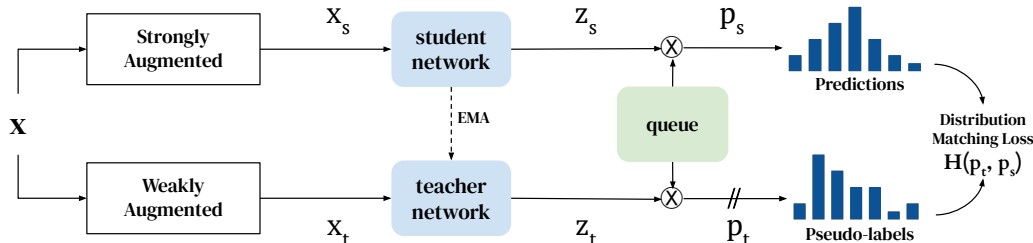

Figure 1: **Q-Match Training Method.** The network is trained by performing continuous self-distillation by minimizing the cross-entropy between student-teach distributions. The two slanted lines denote a stop-gradient operation. The queue is updated with $z_t$ at each training step.

original inputs normalizing them. We found this method works just as well as normalizing the data by calculating the statistics on the full dataset. For categorical features, we one-hot encoded all categorical features. Additionally, we experimented with quantile transformation (Gorishniy et al., 2022) of the inputs. However, this only showed benefits in the Adult dataset. Hence, we do not use quantile transformation for other datasets.

## 3 EXPERIMENTS

**Datasets.** In this work we use the following datasets: Higgs (Baldi et al., 2014), Cover Type Asuncion & Newman (2007), Adult (Kohavi et al., 1996) and MNIST (LeCun et al., 2010). We take various subsets of the original datasets for different experiments. For the exact splits used in different experiments, please refer to Table 8 in the Appendix.

**Implementation.** We implement our method in JAX (Bradbury et al., 2018). The experiments were conducted on single V100 and P100 GPUs.

### 3.1 COMPARISON WITH BASELINES

In this section, we want to measure how our method compares with other self-supervised methods that have been evaluated on tabular data. We find that different papers evaluate their approach on different datasets with different split settings. The pretext set size and labeled set size vary in the original experiments conducted in these papers. To be fair in our comparison to prior work, we report the performance of Q-Match with the same downstream dataset sizes as used by the original authors.

**Higgs Dataset.** First, we compare Q-Match against three other self-supervised methods for tabular data on the Higgs (Baldi et al., 2014) dataset: TabNet (Arik & Pfister, 2021), i-Mix (Lee et al., 2020), and CORE (Han & Ranganath, 2021). We report the results of this experiments in Table 1. Q-Match outperforms all the baselines under all the different splits. In particular, we observe that Q-Match outperforms TabNet with a pretext dataset size that is 100 times smaller.

**Cover Type Dataset.** Next, we compare the performance of our method with baselines on the Cover Type dataset (Asuncion & Newman, 2007). We report the results of our experiment in Table 2. There are two commonly used splits: Cover Type 10% and Cover Type 15k. In the fine-tuning setup, we observe that Q-Match outperforms Contrastive MixUp by a margin of about 10%. We hypothesize this gain in performance is due to the fact that in the contrastive loss the encoder mistakenly considers samples belonging to the same class as negatives. The Q-Match loss does not suffer from this problem. The encoder is free to learn the similarities between different samples and does not consider all items in the batch as negatives. In the linear evaluation setup, we observe Q-Match is about 0.7% to 1.6% better than two versions of i-Mix that use the contrastive loss. Q-Match is slightly worse (0.3%) than i-Mix BYOL in the linear evaluation setup.

**MNIST 10% Dataset.** Next, we perform a similar comparison on the MNIST (LeCun et al., 2010) dataset. While it is an image dataset, prior work (Yoon et al. (2020); Darabi et al. (2021) has used MNIST 10% for research in tabular domain by converting the pixels in an image into a flattened vector. In the past, only 10% of the training set is used as labeled data while the rest 90% of the training set is used as pretext data. We report the results of this experiment in Table 3. We observe

| Method | Pretext Dataset Size | Labeled Dataset Size | Accuracy |
|---|---|---|---|
| CORE (Han & Ranganath, 2021) | 50k | 5k | $66.92 \pm 0.55$ |
| Q-Match (Ours) | 50k | 5k | $\mathbf{68.13 \pm 1.02}$ |
| TabNet (Arik & Pfister, 2021) | 10M | 10k | $68.96 \pm 0.39$ |
| Q-Match (Ours) | 10M | 10k | $\mathbf{71.13 \pm 0.21}$ |
| TabNet (Arik & Pfister, 2021) | 10M | 100k | $73.19 \pm 0.15$ |
| i-Mix (Lee et al., 2020) | 100k | 100k | $72.9$ |
| Q-Match (Ours) | 100k | 100k | $\mathbf{73.27 \pm 0.19}$ |

Table 1: **Higgs experiments.** CORE, TabNet, and Q-Match all report fine tuning accuracy, while i-Mix reports the linear classification accuracy.

that while Q-Match outperforms VIME by a margin of about 2%, our method is comparable in performance with the Contrastive Mixup method without using the MixUp augmentation.

| Method | Cover Type 10% Accuracy | Cover Type 15k Accuracy |
|---|---|---|
| Constrastive MixUp (Darabi et al., 2021) | $80.41 \pm .205$ | - |
| $i$-Mix N-Pair (Lee et al., 2020) | - | $72.1 \pm 0.2$ |
| $i$-Mix MoCo v2 (Lee et al., 2020) | - | $73.1 \pm 0.1$ |
| $i$-Mix BYOL (Lee et al., 2020) | - | $\mathbf{74.1 \pm 0.2}$ |
| Q-Match Finetune (Ours) | $\mathbf{90.26 \pm .11}$ | $82.76 \pm .07$ |
| Q-Match Linear (Ours) | - | $73.79 \pm .22$ |

Table 2: **Cover Type experiments.** For Cover Type 15k, i-Mix uses a linear classifier on top of the pretext features. For Constrastive MixUp, the evaluation uses fine tuning. We present results of Q-Match for both tasks.

| Method | Accuracy |
|---|---|
| VIME (Yoon et al., 2020) | $95.77 \pm .22$ |
| Constrastive MixUp (Darabi et al., 2021) | $97.58 \pm .08$ |
| Q-Match (Ours) | $\mathbf{97.67 \pm .21}$ |

Table 3: **MNIST 10% experiments.** Fine tuning accuracy for VIME, Constrastive MixUp, and Q-Match.

## 3.2 Data Scaling Experiments

In the previous subsection, we show that Q-Match either outperforms or is at par with the other self-supervised baselines when the pretext size is large and all the samples in the labeled downstream dataset are used for evaluation. In this experiment, we want to measure how these methods perform if the amount of labeled data and the pretext dataset size changes. To compare fairly against all the baselines, for this set of experiments, we re-implement the following methods: TabNet, VIME, and i-Mix (N-Pair version).

**Training Details.** In all the experiments in this section, we first train the encoder on the pretext task. We then proceed to the downstream tasks which can either be fine-tuning or linear evaluation. We always compare against the supervised learning baseline. This baseline refers to the method where we do not train a model with any pretext task but begin downstream task training from random initialization. We perform a grid search over relevant hyper parameters for each method. For all methods, we search over pretext task and downstream task learning rates. For the TabNet, VIME, SimCLR, SimSiam, DINO, VICReg, and Q-Match algorithms, we added the probability of corrupting a column (as defined in Yoon et al. (2020)) to the grid. For the N-Pair i-Mix algorithm, we also add the loss temperature to the grid. For the Q-Match algorithm, we also add the queue size and the student temperature to the grid. Please refer to Table 6 in the Appendix for more details on the parameters. We picked the best parameters according to the validation dataset for each downstream task. For all experiments, we report the average and standard deviation over 5 trials. We train all

pretext tasks using a maximum of 200 epochs with early stopping and a patience of 32 (using the pretext validation dataset). For the supervised tasks, we train for a maximum of 500 epochs with a patience of 32 on the validation accuracy. All tasks use a batch size of 512 and the parameters are updated using Adam citepadamw optimizer with weight decay. All algorithms used zero weight decay during pretraining and $10^{-1}$ weight decay during the downstream tasks.

**Corruption Function.** One important factor that affects the performance is the choice of the corruption function used to create the two augmented views. Both VIME and TabNet augment the orignal data by randomly corrupting columns. The VIME corruption function replaces corrupted values with samples from the pretext dataset while the TabNet corruption function replaces values with zeros. We always use the VIME corruption function in all our experiments. Even for our implementation of the TabNet algorithm, we use the VIME corruption function as we found the VIME corruption performed better. Unless otherwise stated, we do not corrupt the teacher view and only corrupt the student view.

**Few Shot Learning.** In this experiment, we compare the representations learned by different learning algorithms by only using 1% of the total available labels for evaluation. This scenario is common in industry when number of available labels for a downstream task might be less but there might a lot of unlabeled data available to learn an encoder. In addition to the tabular self-supervised algorithms we used as comparisons for other experiments, we also include results for our implementations of the following methods that have previously been used for image self-supervision tasks: SimCLR (Chen et al., 2020b), SimSiam (Chen & He, 2021), DINO (Caron et al., 2021), and VICReg (Bardes et al., 2021). For SimCLR, we also include a large batchsize (4096) version, since this has been found to be helpful (Chen et al., 2020b). For all datasets, we only use one-percent of the original labels for downstream training and keep the rest of the data for pretext training and validation. We report the performance of learning a linear classifier in Table 5 and fine-tuning the entire encoder in Table 4.For the Linear Classification Task, Q-Match outperforms all other methods on all datasets except VICReg on MNIST, which it performed similarly. For the Finetuning task, Q-Match was competitive on all datasets, but did especially well on the Cove rType and Higgs datasets, where it had the top accuracies.

|  | **Dataset** | | | |
| **Algorithm** | Cover Type 1% | Higgs100k 1% | Adult 1% | MNIST 1% |
| Supervised Baseline | $70.33 \pm 1.45$ | $56.29 \pm 2.58$ | $78.14 \pm 0.70$ | $88.81 \pm 0.34$ |
| VIME | $71.47 \pm 0.25$ | $62.90 \pm 2.29$ | $79.70 \pm 1.81$ | $92.85 \pm 0.48$ |
| TabNet | $70.34 \pm 0.88$ | $62.12 \pm 1.3$ | $78.18 \pm 1.17$ | $86.94 \pm 1.88$ |
| i-Mix | $71.45 \pm 0.23$ | $59.63 \pm 2.79$ | $77.66 \pm 1.50$ | $92.12 \pm 0.42$ |
| SimCLR | $71.80 \pm 0.41$ | $64.26 \pm 0.51$ | $80.32 \pm 1.47$ | $94.67 \pm 0.36$ |
| SimCLR (large batch) | $71.86 \pm 0.19$ | $63.14 \pm 0.66$ | $80.13 \pm 1.01$ | $93.97 \pm 0.19$ |
| SimSiam | $71.95 \pm 0.32$ | $58.42 \pm 1.28$ | $78.45 \pm 3.38$ | $93.03 \pm 0.71$ |
| VICReg | $69.92 \pm 1.47$ | $61.15 \pm 3.80$ | $\mathbf{80.62 \pm 0.42}$ | $\mathbf{97.40 \pm 0.09}$ |
| DINO | $72.58 \pm 0.54$ | $56.69 \pm 4.11$ | $78.35 \pm 0.81$ | $90.10 \pm 1.20$ |
| Q-Match (Ours) | $\mathbf{72.69 \pm 0.34}$ | $\mathbf{65.19 \pm 1.25}$ | $79.05 \pm 2.80$ | $96.32 \pm 0.54$ |

Table 4: Accuracy of our method versus other pretext algorithms on the finetuning classification task.

**Varying Downstream Dataset Size.** In this experiment we want to measure how downstream task performance changes as more labels are available. We increase the fraction of labeled data available in the Higgs dataset and train the downstream task. Note that in this experiment the pretext size is fixed for all methods at 100k samples. We report the results of this experiment in Figure 2. We observe that Q-Match outperforms other methods across different fractions of labeled data in both the fine-tuning and linear evaluation tasks. The difference in performance between and other methods increases as the number of labeled samples becomes less. In other words, Q-Match is more sample efficient in terms of labels required. Also note that in the low labeled data regime, self-supervised pre-training using Q-Match provides a good initialization for the downstream task. This initialization leads to about 10% improvement in performance than simply performing supervised learning from random initialization.

**Varying Pretext Dataset Size.** In this experiment we want to measure how downstream task performance changes as more unlabeled data is available for the pretext task training. We increase the fraction of unlabeled data available in the Higgs dataset and run both the pretext training and

| | Dataset | | | |
|---|---|---|---|---|
| **Algorithm** | Cover Type 1% | Higgs100k 1% | Adult 1% | MNIST 1% |
| Supervised Baseline | $70.18 \pm 0.20$ | $60.35 \pm 0.34$ | $78.54 \pm 0.27$ | $85.77 \pm 0.22$ |
| VIME | $68.55 \pm 0.32$ | $64.36 \pm 0.62$ | $76.87 \pm 3.04$ | $87.48 \pm 0.34$ |
| TabNet | $48.07 \pm 7.04$ | $60.57 \pm 0.69$ | $76.83 \pm 1.08$ | $23.42 \pm 8.48$ |
| i-Mix | $67.52 \pm 0.39$ | $60.05 \pm 0.5$ | $75.62 \pm 0.57$ | $90.58 \pm 0.21$ |
| SimCLR | $69.69 \pm 0.19$ | $65.22 \pm 0.83$ | $79.83 \pm 0.38$ | $92.26 \pm 0.39$ |
| SimCLR (large batch) | $70.13 \pm 0.20$ | $65.64 \pm 0.92$ | $77.45 \pm 2.87$ | $92.70 \pm 0.48$ |
| SimSiam | $64.66 \pm 1.43$ | $56.60 \pm 3.68$ | $79.27 \pm 1.95$ | $92.21 \pm 2.39$ |
| VICReg | $65.72 \pm 1.53$ | $65.79 \pm 0.22$ | $76.88 \pm 1.49$ | $\mathbf{97.52 \pm 0.06}$ |
| DINO | $61.47 \pm 3.08$ | $57.17 \pm 2.49$ | $77.11 \pm 0.99$ | $67.94 \pm 5.21$ |
| Q-Match (Ours) | $\mathbf{70.39 \pm 0.59}$ | $\mathbf{67.22 \pm 0.29}$ | $\mathbf{80.53 \pm 0.38}$ | $97.11 \pm 0.47$ |

Table 5: Linear classification accuracy of our method versus other pretext algorithms.

downstream task training. Note that in this experiment the downstream labeled set size is fixed for all methods at 100k samples. We report the results of this experiment in Figure 3. We observe that Q-Match outperforms other methods. The difference in performance is especially stark when fewer samples are available. In other words, Q-Match is more sample efficient even in terms of unlabeled data requirements. Depending on the amount of unlabeled data available, Q-Match can increase the performance on downstream task by 5%-8% in the finetuning setup.

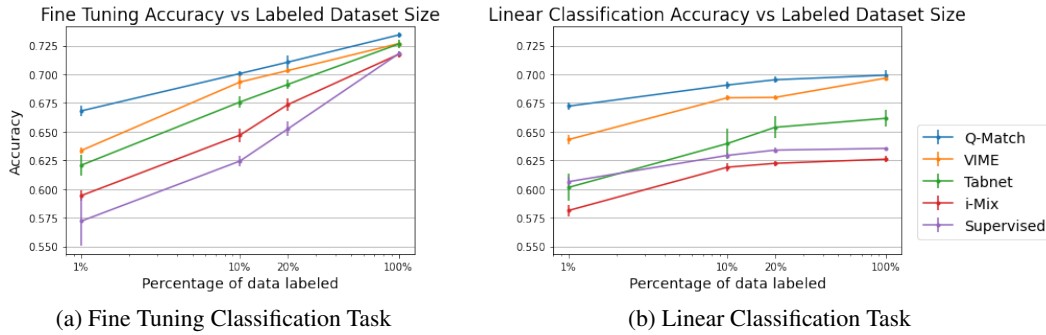

(a) Fine Tuning Classification Task

(b) Linear Classification Task

Figure 2: Classification performance as the size of the labeled data increases for the Higgs100k dataset.

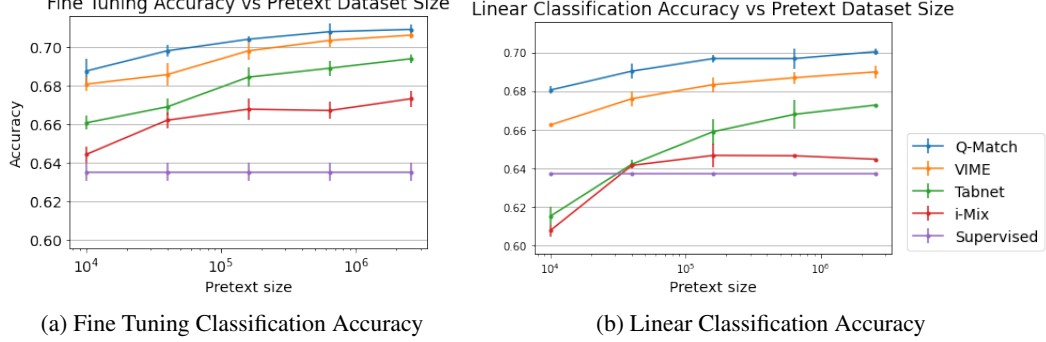

(a) Fine Tuning Classification Accuracy

(b) Linear Classification Accuracy

Figure 3: Accuracy for different sizes of the pretext set for the Higgs dataset.

## 3.3 SENSITIVITY ANALYSIS

In this subsection, we study how some hyperparameter choices affect downstream task performance. In particular, we find our method's final performance is affected significantly by the corruption probability and the queue size. Below we report how sensitive the learning algorithm is for these datasets: Higgs and Cover Type.

**Corruption Probability.** We experiment varying the corruption probability in both views in Q-Match. The linear classification results are shown in Figure 4. The more yellow/bright the color of the grid, the better the performance while more blue/dark means the performance is worse. Note that in both the Cover Type and the Higgs experiments, there appears be both a maximum teacher and student corruption probability that is beneficial for pre-training with Q-Match. After this maximum value, there is too much corruption of the original input to learn a useful representation. Additionally, there is also a critical maximum value of the sum of both probabilities going beyond which, the model fails to learn useful representations using Q-Match. In other words if both the student and teacher are corrupted with high corruption probability, it leads to sub-optimal performance. *We recommend using a small or zero value for the teacher corruption, and a moderate value for the student corruption to achieve optimal downstream performance.*

On the other hand, we find the fine-tuning results to be fairly robust to these parameters. The initialization found by performing Q-Match with different values of corruption probability is good enough for downstream fine-tuning to achieve optimal performance.

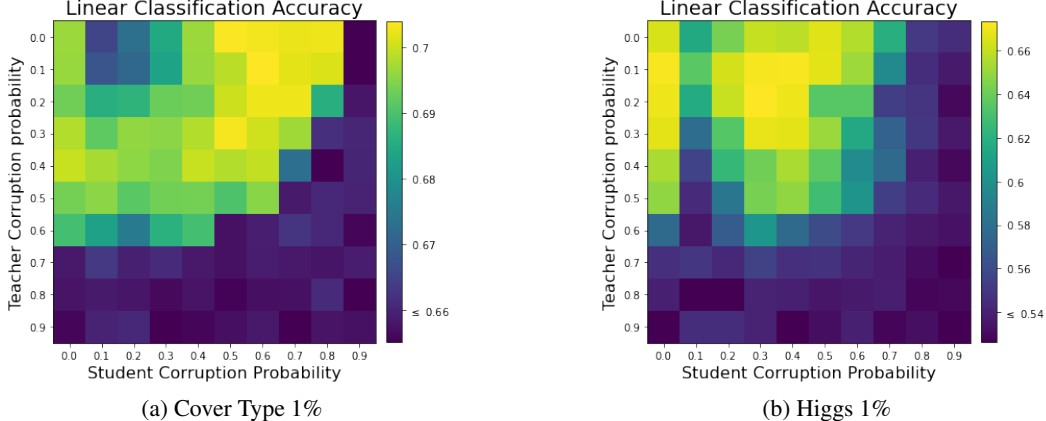

(a) Cover Type 1%                                                    (b) Higgs 1%

Figure 4: Linear classification accuracy as corruption of the features changes for both the student and the teacher views.

**Queue size.** We examine the effect of changing the size of the queue used in Q-Match for the Higgs, Cover Type, and MNIST datasets. The linear classification performance (in the few shot setting) as a function of the queue size is shown in Figure 5. First, we observe that when the queue size becomes very small, the final performance declines. Second, we note as the queue size increases, there is the less variance in the downstream results. *We recommend using a larger queue ($\geq 10^3$) to achieve optimal performance.*

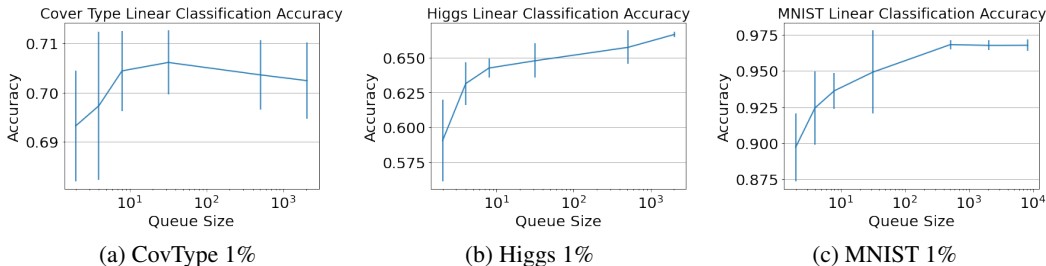

Figure 5: Effect of the queue size on the linear classification accuracy.

## 4 DISCUSSION AND RELATED WORK

**Contrastive Self-supervised Learning.** In the context of self-supervised learning, a class of approaches that have been effective are built on top of the InfoNCE loss (Oord et al., 2018) where for any data point $x_i$ and its corresponding embedding $z_i = f(x_i, \theta)$ there exists a set of positives $P_i$ and negatives $N_i$, loss $L_i^{\text{InfoNCE}}$ is defined as follows:

$$\mathcal{L}_i^{\text{InfoNCE}} = -\log \frac{\sum_{z^+ \in \mathcal{P}_i} \exp\left(z_i \cdot z^+/\tau\right)}{\sum_{z^+ \in \mathcal{P}_i} \exp\left(z_i \cdot z^+/\tau\right) + \sum_{z^- \in \mathcal{N}_i} \exp\left(z_i \cdot z^-/\tau\right)} \tag{1}$$

where $(z_i, z_i^+)$ are positive pairs, $(z_i, z^-)$ are negative pairs and $\tau$ is the softmax temperature. This loss aims to attract the positives closer to each other while repelling the negatives farther from each other. In SimCLR (Chen et al., 2020a), the positives are two views of the same data and negatives are all the other elements in the current mini-batch. In MoCo (He et al., 2020), the positives are the same as SimCLR but the negatives are derived from a queue of past embeddings produced by the model.

Both Contrastive Mixup and i-Mix - N - Pair use the InfoNCE loss. We find Q-Match outperforms both these methods consistently. This could be due to the fact the contrastive loss mistakenly considers samples from the same class as negatives. For a dataset whose classes are uniformly distributed, the probability that at least one sample in the batch of negatives belongs to the same downstream class as the positives is equal to $1 - \left(\frac{N-1}{N}\right)^{(B-1)}$ where $N$ is the number of classes in the downstream task and $B$ is the batch size used during training. This means the occurrence of this event is less in a large-scale and diverse image datasets like ImageNet. Concretely, with a batch size of 512 and the number of downstream classes equal to 1000, there is a probability of approximately 0.4 that an element of the same class will be considered a negative. However in the tabular setting where the number of classes in the downstream task is usually much less (say 10), the probability of considering items of the same class as negatives is much higher (very close to 1). We hypothesize that this might be the reason that using the vanilla N-pair contrastive loss usually results in sub-par performance.

**Non-Contrastive Self-supervised Learning.** In order to remove the dependency on explicit negatives during training, researchers have proposed two types of losses. The first class of approaches like BYOL(Grill et al., 2020), SimSiam (Chen & He, 2021) aim to directly minimize the distance between the positive embeddings using a Mean Squared Error (MSE) loss.

$$\mathcal{L}_i^{\text{MSE}} = -z_i \cdot z_i^+ \tag{2}$$

In BYOL (Grill et al., 2020), $z^+$ is derived from a different view that is passed through a momentum encoder while in SimSiam $z^+$ comes from a different view passed through the same encoder except it has a stop-grad operation to prevent embedding collapse. We compare Q-Match with i-Mix BYOL on the Cover Type dataset in Table 2 and find it is competitive with our method. But our implementation of i-Mix BYOL failed to achieve good performance. We leave exploring i-Mix BYOL on low labeled data regime as future work.

The second class of losses that do not require explicit negatives use prototypes to minimize the cross-entropy between two distributions induced by the positive pair. Examples of this approach are SwAV (Caron et al., 2020) and DINO (**?**).

$$\mathcal{L}_i^{\text{DINO}} = H\left(\frac{z_i^+ \cdot P}{\tau}, \frac{z_i \cdot P}{\tau}\right) \tag{3}$$

where $P$ is a list of learnable prototypes maintained by the model. To avoid embedding collapse DINO uses momentum encoding with a combination of centering and sharpening while SwAV uses Sinkhorn clustering.

Like DINO and SWAV, Q-Match also uses the cross-entropy between two distributions to learn an encoder. But Q-Match differs from them in two aspects. First, while DINO and SWAV use learnable prototypes to induce the target distribution, we use a queue to produce the target distribution. Second, instead of relying on centering and sharpening or Sinkhorn clustering, we use a queue of past embeddings to prevent embedding collapse. Since the queue of embeddings keeps getting refreshed, it is non-trivial for the model to collapse.

**Reconstruction-based Self-supervised Learning.** A commonly used approach for self-supervised learning is using a reconstruction loss to solve a de-noising pretext task. Methods, like VIME (Yoon et al., 2020) and (Arik & Pfister, 2021) take the original sample in the data, and corrupt some of its values using a corruption mask sampled from a Bernoulli distribution. They then learn an encoding function as well as a reconstruction function which aims to reconstruct the original sample using both the learned parameters of the encoder and the parameters specific to the reconstruction task. TabNet is pre-trained to only predict the reconstructed values. On the other hand, VIME also proposes predicting the corruption mask in addition to predicting the reconstructed values. We find the corruption function introduced in these papers useful for producing student and teacher views in Q-Match. However, we find the distribution matching loss to be more effective in the low data regime than the reconstruction based losses.

**Semi-supervised Learning** Our approach is also closely related to the semi-supervised approaches like FixMatch (Sohn et al., 2020) and PAWS (Assran et al., 2021) that learn encoders by matching student-teacher distributions. The difference is that we focus on the self-supervised setup where the downstream task fine-tuning happens after the self-supervised pre-training stage. Hence, we cannot assume knowledge of any known classes during the pre-training stage. Instead, we use a queue to induce the student and teacher distributions. Additionally, we study the problem in the context of tabular datasets while the above mentioned papers are evaluated on image datasets.

## 5 CONCLUSION

We introduced a new self-supervised algorithm, Q-Match, that learns new, useful representations of tabular data entirely from unlabeled data. Q-Match utilizes a queue to perform continuous self-distillation by matching the student distribution to the teacher distribution as training proceeds. We show that Q-Match outperforms existing self-supervised algorithms and supervised learning on tabular datasets in terms of classification accuracy. Additionally, we show that Q-Match is more efficient than existing methods in terms of sizes of the unlabeled pretext dataset and of the labeled downstream dataset. Q-Match continues to outperform other methods when the size of both the pretext and the downstream datasets is increased.

## 6 BROADER IMPACT

Q-Match has minimal assumption about the distribution or the domain of data. Hence, it can be applied to data that is not tabular or a mix of tabular and non-tabular domains as well. We show that Q-Match reduces the amount of unlabeled pretext data and labeled downstream data. This can be advantageous in domains when collecting data itself is expensive, when samples can only be labeled after waiting a long time, or even when the total number of possible labels are naturally limited (e.g., rare medical diagnoses).

On the other hand, even if the encoder is learned in a self-supervised manner without labels it might be biased against certain groups, especially if they are rarely represented in the data. It is recommended to visualize embeddings to identify such cases. Performing detailed group-wise analysis on the downstream task metric will also surface any bias learned by the encoder.

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

## A    APPENDIX

### A.1    ALGORITHM

In Algorithm 1 we show the pseudo code of our method.

### A.2    HYPER PARAMETERS

In Table 6 we show the values of hyper-parameters we search over. For each method we perform a grid search over its relevant hyperparameter values and choose the best one. We use a value of $\tau_{EMA} = 0.9$ for the momentum encoder and a teacher temperature, $\tau_t$, of 0.04 in Q-Match.

**Algorithm 1** Pseudo Code for a Single Q-Match Training Step

```
# Create two views of the same data.
teacher_view = aug(x, teacher_corruption)
student_view = aug(x, student_corruption)

# Pass views through the model.
teacher_embed = encoder(teacher_view, ema_params)
student_embed = encoder(student_view, params)

# Normalize embeddings.
teacher_embed_norm = stop_gradient(l2_normalize(teacher_embed))
student_embed_norm = l2_normalize(student_embed)

# Compute student and teacher distributions.
teacher_logits = teacher_embed_norm @ Q.T / teacher_temperature
teacher_dist = softmax(teacher_logits)
student_logits = student_embed_norm @ Q.T / student_temperature
student_dist = softmax(student_logits)

# Compute loss.
loss = cross_entropy(teacher_dist, student_dist)
gradients = compute_gradients(loss, params)

# Update params, ema params, and queue.
params = update_params(gradients, params)
ema_params = tau * ema_params + (1 - tau) * params
Q = update_queue(Q, teacher_embed_norm)
```

| Parameter Name | Search Space | Relevant Algorithm |
|---|:---:|---|
| Learning rate | $[10^{-5}, 10^{-4}, 10^{-3}, 10^{-2}]$ | All |
| Pre-text learning rate | $[10^{-5}, 10^{-4}, 10^{-3}]$ | All |
| Temperature, $\tau$ | $[0.04, 0.10, 0.15, 0.20, .30]$ | i-Mix |
| Corruption probability | $[.3, .4, .5]$ | VIME, TabNet, SimCLR, SimSiam, VICReg, DINO |
| Student corruption | $[.3, .4, .5]$ | Q-Match |
| Student temperature, $\tau_s$ | $[0.05, 0.1, 0.2]$ | Q-Match |
| Queue size | $[2^9, 2^{11}]$ | Q-Match |

Table 6: Hyper parameter spaces for all algorithms used during training.

## A.3 IMAGENET EXPERIMENTS

As our proposed approach is domain-agnostic, it can also be used to learn image features in a self-supervised manner. We pre-train a ResNet-50 model with the Q-Match loss. We use the multi-crop training setup of NNCLR Dwibedi et al. (2021) and SWAV Caron et al. (2020) where 2 crops of $224 \times 224$ and 6 crops of $96 \times 96$ are used in a single batch. The two larger sized images serve as the teacher views. We pre-train the model for 800 epochs. We use a student temperature of 0.1 and teacher temperature of 0.04. We do not use any color jitter augmentation on the teachers' view to produce the weakly augmented view. Crop augmentation is used on both views. We use a queue size of 98304, embedding size of 256, momentum of 0.99, and the same projection MLP used in NNCLR.

We report the results of linear evaluation of the 2048-d output from the ResNet-50 model in Table 7. We find that training with the Q-Match loss results in better performance than training with the baseline methods. In particular, the performance improvement over DINO is interesting because both DINO and Q-Match use the student-teacher distribution matching loss. While DINO uses learnable prototypes to induce these distributions, Q-Match uses a queue of past embeddings. This further validates the utility of the queue in the student-teacher distribution matching framework. Furthermore, even though Q-Match was developed primarily with tabular datasets with a low number of downstream classes, it is capable of strong performance on image datasets with a larger number of classes.

| Method | Accuracy |
|---|---|
| SWAV (Caron et al., 2020) | 75.3 |
| DINO (Caron et al., 2021) | 75.3 |
| NNCLR (Dwibedi et al., 2021) | 75.6 |
| Q-Match (Ours) | **76.0** |

Table 7: **ImageNet Linear Evaluation.** Linear evaluation accuracy for NNCLR, SWAV, and Q-Match.

| Dataset Name | Pretext Training Set Size | Downstream Training Set Size | Downstream Test Set Size | Experiments |
|---|---|---|---|---|
| Higgs 1% | 98k | 980 | 500k | Few Shot, Sensitivity |
| Higgs 5k | 50k | 5k | 25k | Baseline |
| Higgs 10k | 10M | 10k | 500k | Baseline |
| Higgs 100k | 100k | 100k | 500k | Baseline |
| Higgs Variable Labeled | 98k | {980, 9.80k, 19.6k, 98k} | 500k | Data Scaling |
| Higgs Variable Pretext | {10k, 40k, 160k, 640k, 2.56M} | 10k | 500k | Data Scaling |
| Cover Type 1% | 113400 | 1134 | 429812 | Few Shot, Sensitivity |
| Cover Type 10% | 464809 | 46480 | 116203 | Baseline |
| Cover Type 15k | 11340 | 11340 | 565892 | Baseline |
| Adult 1% | 8170 | 86 | 16281 | Few Shot |
| MNIST 1% | 57k | 600 | 10k | Few Shot, Sensitivity |
| MNIST 10% | 60k | 10k | 10k | Baseline |

Table 8: Dataset splits.

## A.4 DATASETS AND SPLITS

In Table 8, we show in detail the datasets and their splits. Different experiments required us to create different splits. For comparison with baseline methods, we attempt to stick to the splits used in the baseline papers. For data scaling and few-shot experiments, we create our own splits and train all methods on the same splits.

