# OpenReview forum: "Q-Match: Self-Supervised Learning For Tabular Data by Matching Distributions Induced by a Queue"
_ICLR.cc/2023/Conference — Submitted to ICLR 2023_

### Official Review · Reviewer_VckF · 2022-10-27

**Confidence:** 4
**Clarity, Quality, Novelty And Reproducibility:** This paper lacks of originality.
**Correctness:** 3
**Technical Novelty And Significance:** 1
**Empirical Novelty And Significance:** 2
**Recommendation:** 3

**Strength And Weaknesses:**

Strength:

This paper has conducted some interesting experiments.

Weaknesses:

1. The technical novelty of this paper is very limited. The proposed method is just a combination of some existing methods.

2. The motivation of the proposed method is unclear. This paper has not explained why the proposed method can achieve proper performance.

3. The writing of this paper requires significantly improvement. There are many typos and grammar mistakes. For example, in the **method** subsection of Chapter 2, the word "student" in the sentence "We pass $x_{i,s}$ Through the student model to produce the student embedding $z_{i,s}$" should be "teacher", In the **Semi-supervised Learning** subsection of Charpter 4, the word "teach" in the second last sentence may be "teacher".

**Summary Of The Paper:**

This paper proposes to use a queue of embeddings of samples from the unlabeled dataset to improve the performance of self-supervised learning. It presents that the proposed method have achieved success on the tabular datasets.

**Summary Of The Review:**

The technical novelty of this paper is very limited.

---

> ### Author Response · Authors · 2022-11-12
> **Authors’ response to reviewer VckF**
>
> We thank the reviewer for their time and effort. We address the concerns of the reviewer below.
>
>
>
> 1. _“The technical novelty of this paper is very limited. The proposed method is just a combination of some existing methods.”_
>
> Our proposed approach has certain similarities with existing methods which we describe in the related work and in the experiments sections. The main similarity with FixMatch is that we are using teacher and student distribution matching. However it is impossible to apply the FixMatch framework directly in the self-supervised setup because it requires knowledge of the underlying classes. We propose to use a queue to induce these distributions. To the best of our knowledge,using a queue for this purpose is novel.
>
> The closest method to our approach in the self-supervised setup would be DINO. It also proposes to use teacher and student distribution matching but uses learned prototypes to induce the distributions. With learned prototypes, the training results in the trivial solution of the model assigning the same prototype to all embeddings. To alleviate this and achieve strong performance, DINO uses a combination of weight normalization, centering, sharpening, temperature and weight scheduling. We show how using a queue to induce these distributions instead of learned prototypes can be a better and arguably simpler approach.
>
> Additionally, in the context of tabular data, we present new experimental comparisons with many different self-supervised learning methods. Please also consider the comparison made with methods such as SimSiam, VICReg, We believe such extensive exploration of self-supervised losses has not been done yet on tabular data. Furthermore we show how this method developed on tabular data can achieve 76% ImageNet linear evaluation accuracy.
>
> 2. _“The motivation of the proposed method is unclear. This paper has not explained why the proposed method can achieve proper performance.”_
>
> We thank the reviewer for the feedback and will attempt to make the motivation clearer in the paper. The main point of motivation is the FixMatch paper which shows one can use an unlabeled dataset to improve performance by using student-teacher distribution matching. **However, one cannot apply FixMatch directly without knowing the classes beforehand**. Therefore, we propose to use a queue of embeddings (which come from unlabeled data) instead of classes. The idea here is that the class distribution will be subsumed in the distribution induced by the queue, as there will be elements of different classes in the queue. With this motivation, we experiment and find out that indeed this method results in an increase in performance over multiple datasets especially when the number of labeled samples is less. There are multiple reasons why this method performs better:
> i) Our method does not use any explicit negatives like SimCLR. In SimCLR, all the other elements in a training batch are negatives. We present a theoretical reason (see Section 4: Discussion and Related Work) why SimCLR works really well on image datasets with a large number of underlying classes. But when the number of underlying classes are less (as is the case with tabular datasets), the requirement of having a large batch size works against SimCLR as it treats elements of the same underlying class as negatives while learning representations. Our method alleviates the reliance on explicit positives/negatives and results in strong performance irrespective of the number of classes. We show our method achieves state of the art performance on a number of tabular datasets (with less number of underlying classes) and image datasets (with a large number of underlying classes).
> ii) By using a queue that is updated every time, the model has more difficulty in overfitting or collapsing to a trivial solution.
>
> 3. _“The writing of this paper requires significantly improvement.”_
>
> We thank the reviewer for pointing the typos out. We will fix these mistakes. We will go through the paper again and remove any other typos we find.

---

### Official Review · Reviewer_CX2D · 2022-10-29

**Confidence:** 4
**Correctness:** 3
**Technical Novelty And Significance:** 1
**Empirical Novelty And Significance:** 1
**Recommendation:** 3

**Clarity, Quality, Novelty And Reproducibility:**

Providing the pseudo-code (Algorithm 1) is insufficient to reproduce many experimental results in the article.

**Details Of Ethics Concerns:**

N.A.

**Strength And Weaknesses:**

Strength:

1. The paper is relatively well-written and easy to follow.
2.  The paper conducts extensive experiments to validate the proposed method empirically.

Weaknesses.

1. The work has limited novelty on top of Fixmatch [Sohn et al., 2020].

**Summary Of The Paper:**

This paper adapts the Fixmatch method [Sohn et al., 2020] from semi-supervised to self-supervised learning, and significantly improves the results compared with the benchmarks in experiments.

**Summary Of The Review:**

The contribution of this article is insufficient to be accepted to the conference. However,  I believe the paper is an excellent workshop paper.


###########

I thank the authors for their response. However, I still think the contribution of the article is relatively small so keep my score unchanged.

---

> ### Author Response · Authors · 2022-11-12
> **Authors’ response to reviewer CX2D**
>
> We thank the reviewer for acknowledging our writing and the extensive experiments in our work.  We address their concerns below.
>
> 1. _“The work has limited novelty on top of Fixmatch [Sohn et al., 2020]”_
>
> We agree with the reviewer regarding our approach adapting FixMatch’s student-teacher distribution matching framework in the self-supervised setup. **However, it is not fair to say that the paper has** “ limited novelty on top of Fixmatch.” This is because one **cannot apply FixMatch directly without knowing the classes beforehand**. Our paper proposes a simple approach to circumvent this limitation by using a queue of embeddings. Furthermore, we show how our approach outperforms many other self-supervised methods on a number of tabular datasets. We also find the proposed approach is applicable in the image domain and outperforms existing methods (achieves 76% linear evaluation accuracy on ImageNet). We ask the reviewer to reconsider their remark as the FixMatch paper does not handle self-supervised scenario.
>
> 2. _“Providing the pseudo-code (Algorithm 1) is insufficient to reproduce many experimental results in the article.”_
>
> Thank you for recognizing the large number of experiments in our work.  We appreciate your concern for reproducing our work. We will release the code after the review period to facilitate reproducibility. We provide the pseudocode for easy comparison with baseline methods which also include the pseudo-code in their respective papers.
>
>
> 3. _“Some of the paper’s claims have minor issues. A few statements are not well-supported, or require small changes to be made correct.”_
>
> Thanks for the feedback. Please feel free to flag any specific comments, and we will address them.

---

### Official Review · Reviewer_gtA5 · 2022-10-30

**Confidence:** 3
**Correctness:** 4
**Technical Novelty And Significance:** 3
**Empirical Novelty And Significance:** 3
**Recommendation:** 6

**Clarity, Quality, Novelty And Reproducibility:**

The paper is clearly written. I believe the method can be reproduced based on the information provided in the paper.

**Strength And Weaknesses:**

 Strengths:
1.	The paper is well-written and easy to follow. The algorithm is described clearly. For each part of the algorithm, the authors provide the motivation and mention the counterpart in vision methods.
2.	The proposed method is simple and has good performance. The method does not include many tricks, but a rather original implementation of the principled self-supervised learning method. It is encouraging to see it works well on tabular data, across various datasets and settings.
Weaknesses:
1.	Despite the good performance, I’m also curious to see the contribution of each part of the model. How does loss function affect the performance? What if you use the VCIReg loss? How does the queue affect the performance? If you remove the queue and use large batchsize following SimCLR, or simply use Simsiam, what will happen? It would be better if the authors include them as an ablation study.


**Summary Of The Paper:**

This paper proposes Q-match, a self-supervised method for tabular data. Similar to Fixmatch and DINO, the proposed method matches the feature distribution induced by a teach network and a student network, with a feature queue similar to moco. Results on several tabular data benchmarks indicate that the proposed method outperform previous self-supervised methods.

**Summary Of The Review:**

Self-supervised learning has been a very successful technique across vision and NLP. This paper extends the common self-supervised techniques to tabular data and shows its effectiveness. Although the authors could have provided more justifications of their method, I tend to accept this paper.

---

> ### Author Response · Authors · 2022-11-13
> **Authors’ response to reviewer gtA5**
>
> We thank you for recognizing the novelty of the work, the clarity of our writing, and the simplicity of our approach.  We also thank you for offering experiments ideas, many of which we have since implemented.  We ran experiments for SimSiam, VICReg, DINO, and a large batch size SimCLR, and achieved the following results on the few-shot learning experiments:
>
> Fine Tuning Classification Accuracy:
> |                | CoverType        | Higgs            | Adult            | MNIST            |
> |----------------|------------------|------------------|------------------|------------------|
> | DINO           | 0.7258 $\pm$ 0.0054 | 0.5669 $\pm$ 0.0411 | 0.7835 $\pm$ 0.0081 | 0.9010 $\pm$ 0.012   |
> | SimSiam        | 0.7195 $\pm$ 0.0032 | 0.5842 $\pm$ 0.0128 | 0.7845 $\pm$ 0.0338 | 0.9303 $\pm$ 0.0071 |
> | VICReg         | 0.6992 $\pm$ 0.0147 | 0.6115 $\pm$ 0.038  | **0.8062 $\pm$ 0.0042** | **0.9740 $\pm$ 0.0009**  |
> | SimCLR (Large Batch) | 0.7186 $\pm$ 0.0019 | 0.6314 $\pm$ 0.0066 | 0.8013 $\pm$ 0.0101 | 0.9397 $\pm$ 0.0019 |
> | Q-Match (Ours) | **0.7269 $\pm$ 0.0034** | **0.6519 $\pm$ 0.0125** | 0.7905 $\pm$ 0.028  | 0.9632 $\pm$ 0.0054 |
>
> Linear Classification Accuracy:
> |                      | CoverType        | Higgs            | Adult            | MNIST            |
> |----------------------|------------------|------------------|------------------|------------------|
> | DINO                 | 0.6147 $\pm$ 0.0308 | 0.5717 $\pm$ 0.0249 | 0.7711 $\pm$ 0.0099 | 0.6794 $\pm$ 0.0521 |
> | SimSiam              | 0.6466 $\pm$ 0.0143 | 0.566 $\pm$ 0.0368  | 0.7927 $\pm$ 0.0195 | 0.9221 $\pm$ 0.0239 |
> | VICReg               | 0.6572 $\pm$ 0.0153 | 0.6579 $\pm$ 0.0022 | 0.7688 $\pm$ 0.0149 | **0.9752 $\pm$ 0.0006** |
> | SimCLR (Large Batch) | 0.7013 $\pm$ 0.002  | 0.6564 $\pm$ 0.0092 | 0.7745 $\pm$ 0.0287 | 0.927 $\pm$ 0.0048  |
> | Q-Match (Ours)       | **0.7039 $\pm$ 0.0059** | **0.6722 $\pm$ 0.0029** | **0.8053 $\pm$ 0.0038** | 0.9711 $\pm$ 0.0047 |
>
>
>
>
> The image methods in general perform poorly on at least one the tabular dataset tasks.  DINO, SimSiam, and VICReg perform especially poorly on the Higgs dataset for both fine tuning and linear classification.  For the CoverType dataset, the three algorithms vastly underperform on the linear classification task.  Although VICReg seems to perform well for MNIST and the Adult dataset, its numbers were comparable to  Q-Match, and we believe that it underperforming on CoverType and Higgs is evidence that it is better suited for the non-tabular task.  For the large batch size experiments of SimCLR, we used a batch size of 4096.  While it was competitive on many of the datasets and tasks, Q-Match outperformed it on all but the fine tuning classification task on the Adult dataset.  We will include these methods in the final version of the paper.
>
> _“How does queue size affect performance?”_
>
> For the queue size, we included a sensitivity analysis in the original submission (Fig. 5) that shows the accuracy as a function of queue size.  We found that if the queue becomes too small, the performance decreases–we conjecture that this happens because the queue is no longer representative of the population.

---

### Official Review · Reviewer_8ARq · 2022-11-04

**Confidence:** 3
**Correctness:** 4
**Technical Novelty And Significance:** 2
**Empirical Novelty And Significance:** 3
**Recommendation:** 5

**Clarity, Quality, Novelty And Reproducibility:**

The paper is easy to follow. The technical novelty is limited - all building pieces (with slight differences) already exist in other literature and it is similar to FixMatch.

**Strength And Weaknesses:**

Strength
1. The motivation of the proposed method is clear: to build a SSL method similar to the existing semi-supervised methods for tabular data.
2. All the building pieces (with slight differences) exist in SSL for image data and therefore the proposed method is simple and straightforward.

Weakness
1. The novelty is limited to empirical application to tabular data.
2. The paper does not seem to have compared against many similar methods it mentioned (SwAV for instance). As the proposed method is  not tailored for tabular data from a pure methodology perspective, it is unclear if the proposed method makes more sense than just applying DINO, SwAV, etc. to the tabular datasets used in the experiments.

**Summary Of The Paper:**

The paper proposes a self-supervised learning method for tabular data. Specifically, it adopts a pipeline similar to DINO – it matches the categorical distribution produced from an EMA teacher network and a student network but the classifier is built with a queue of past embeddings instead of a learned linear layer. The paper shows empirical results and ablation studies on multiple datasets and demonstrates the effectiveness of the proposed method on tabular data.

**Summary Of The Review:**

The idea of the proposed SSL method is straightforward and seems to be effective. However, it shares too many similarities with SSL methods for images without any pieces which are special to tabular data. It is therefore unclear if the performance gain is unique for the specific design in this paper.

---

> ### Author Response · Authors · 2022-11-12
> **Authors' response to reviewer 8ARq**
>
> _“The novelty is limited to empirical application to tabular data”_ and _“However, it shares too many similarities with SSL methods for images without any pieces which are special to tabular data. It is therefore unclear if the performance gain is unique for the specific design in this paper.”_
>
> We thank the reviewer for pointing out that our method does not have any tabular data specific assumptions. This pushed us to also verify if this loss works well with images. This will also help us compare our approach with existing self-supervised methods. We performed an experiment training a ResNet-50 model with Q-Match on ImageNet dataset in the self-supervised setup. We notice that Q-Match loss can achieve 76.0% linear evaluation accuracy on 800 epoch training with multi-crop. This method currently outperforms NNCLR (75.6%) and SWAV (75.4%).  We used a momentum encoder, and for weak augmentation we did not use any color jitter on the teacher's views.
>
> While we developed the method using tabular data for our experiments, we did not make any hard assumptions about the tabular nature of data. Hence, this method is applicable beyond tabular data. We hope this can convince the reviewers about the utility of the proposed approach in both tabular and non-tabular domains.
>
> _“The paper does not seem to have compared against many similar methods it mentioned (SwAV for instance). As the proposed method is not tailored for tabular data from a pure methodology perspective, it is unclear if the proposed method makes more sense than just applying DINO, SwAV, etc. to the tabular datasets used in the experiments.”_
>
> We thank the reviewer for their advice and for encouraging us to compare against established methods like DINO from the image domain. We also implemented DINO, SimSiam, VICReg and compared them to our method on the few-shot learning experiments–they performed worse than our method on most datasets, especially the tabular datasets (CoverType and Higgs).  We will update our paper to include these new results.
>
>
> Fine Tuning Classification Accuracy
> |                | CoverType        | Higgs            | Adult            | MNIST            |
> |----------------|------------------|------------------|------------------|------------------|
> | DINO           | 0.7258 $\pm$ 0.0054 | 0.5669 $\pm$ 0.0411 | 0.7835 $\pm$ 0.0081 | 0.9010 $\pm$ 0.012   |
> | SimSiam        | 0.7195 $\pm$ 0.0032 | 0.5842 $\pm$ 0.0128 | 0.7845 $\pm$ 0.0338 | 0.9303 $\pm$ 0.0071 |
> | VICReg         | 0.6992 $\pm$ 0.0147 | 0.6115 $\pm$ 0.038  | **0.8062 $\pm$ 0.0042** | **0.9740 $\pm$ 0.0009**  |
> | Q-Match (Ours) | **0.7269 $\pm$ 0.0034** | **0.6519 $\pm$ 0.0125** | 0.7905 $\pm$ 0.028  | 0.9632 $\pm$ 0.0054 |
>
>
> Linear Classification Accuracy
> |                | CoverType        | Higgs            | Adult            | MNIST            |
> |----------------|------------------|------------------|------------------|------------------|
> | DINO           | 0.6147 $\pm$ 0.0308 | 0.5717 $\pm$ 0.0249 | 0.7711 $\pm$ 0.0099 | 0.6794 $\pm$ 0.0521 |
> | SimSiam        | 0.6466 $\pm$ 0.0143 | 0.5660 $\pm$ 0.0368  | 0.7927 $\pm$ 0.0195 | 0.9221 $\pm$ 0.0239 |
> | VICReg         | 0.6572 $\pm$ 0.0153 | 0.6579 $\pm$ 0.0022 | 0.7688 $\pm$ 0.0149 | **0.9752 $\pm$ 0.0006** |
> | Q-Match (Ours) | **0.7039 $\pm$ 0.0059** | **0.6722 $\pm$ 0.0029** | **0.8053 $\pm$ 0.0038** | 0.9711 $\pm$ 0.0047 |

---

> > ### Comment · Reviewer_8ARq · 2022-12-11
> > **Response to the authors**
> >
> > I thank the authors for the efforts. The results show some advantage over the baselines (SwAV, DINO, etc.) on general input modalities. While I have less doubt on the effectiveness of the proposed method, there is still an important missing piece in the story: **why does it outperform these baselines?** As the proposed method share too many similarities with methods like NNCLR, DINO, SwAV, I believe the authors need to provide either some intuition or theoretical analysis on that.
> >
> > Moreover, the story needs to revised with the latest batch of results. The original story is that "the method works well for **tabular data**", but now it seems that there is nothing special with tabular data in neither methodology nor empirical performance. It is hard for me to see how to bridge this gap without rewriting a large portion of the paper.
> >
> > Given these observations, I decide to keep my original rating and encourage the authors to 1) provide a more thorough analysis on why it works and 2) reorganize the whole story.

---

> > > ### Author Response · Authors · 2022-12-13
> > > **Authors' response to reviewer 8ARq**
> > >
> > > We thank the reviewer for replying to our rebuttal.
> > >
> > > We would like to point out that in Discussion section of the paper we attempt to compare Q-Match with other baselines. We clearly point out the similarities and differences of Q-Match with SimCLR, DINO, SwAV, BYOL and VIME. In our reply to reviewer VckF we also mention some more differences with the DINO method.
> > >
> > > We believe the reason for the strong performance is due to the  use of a queue in the student-teacher distribution matching framework. Other methods like DINO and SwAV also rely on this framework have shown strong performance. However, they both rely on learnable prototypes. It is easy for the model to overfit to these prototypes to bring the self-supervised loss down. Techniques like centering, sharpening or Sinkhorn clustering are required to prevent the trivial solution from being learnt. On the other hand, we use a queue that gets refreshed at each step. This prevents the learning of any trivial solutions to bring the self-supervised loss down. We would also like to highlight that while NNCLR and Q-Match both use queues, the way the queues are used are different. NNCLR provides diverse positives by mining the queue. In a single batch only the top N positive samples from the queue are used. On the other hand, Q-Match uses all the elements of the queue to calculate the loss at each step. We can update the paper with this additional intuition.
> > >
> > > Furthermore, we would like to point out that our work provides extensive experiments of 8 self-supervised learning algorithms on different tabular datasets with different amounts of pre-training and downstream data. We believe this will serve the community because we are studying both self-supervised methods developed strictly for tabular data (VIME, i-mix, TabNet) and self-supervised method developed for images (SimCLR, DINO, VICReg) under the same set of experiments. In general, we find that methods developed for images are at par or outperform methods developed specifically for tabular data. This is an important contribution that is consistent with the current storyline.
> > >
> > > We appreciate reviewer 8ARq's concerns that a major rewrite is necessary for the paper. It would be helpful if the reviewer can mention what sections would change. As per the reviewers' request, we can update the paper with more intuition (mentioned above) but that would not constitute a major rewrite. Especially because a majority of the paper is the Proposed Approach and the Experiments sections which would remain unchanged. We would ask the reviewer to reconsider their score given that we showed the method works favourably against many baselines, and even though it was developed using tabular datasets, Q-Match achieves *strong* performance on ImageNet, too.
> > >
> > > We hope the reviewer can consider this context to provide their final score.

---

### Author Response · Authors · 2022-11-19
**General Response**

We thank the reviewers for taking the time and the effort to review our submission.
We would like to emphasize some positive aspects of the paper: the reviewers thought we _conducted interesting experiments_ (VckF), had a _clear motivation of building a self-supervised method similar to the existing semi-supervised methods for tabular data_ (8ARq), _conducted extensive experiments to validate the proposed method empirically_ (CX2D), and _thought our approach was a rather original implementation of the principled self-supervised learning method_ (gtA5).

We address some of the common concerns raised by the reviewers below.

**Novelty on top of FixMatch**

A concern shared by some reviewers (CX2D, VckF) is that our method “has limited novelty on top of FixMatch”. However, we would like to point out that FixMatch cannot be applied in the self supervised setup without knowing the downstream classes during pre-training. On the other hand, our proposed method shows how we can take advantage of the powerful student-teacher distribution matching framework in the self-supervised setup. Additionally, our proposal of using a queue of embeddings to induce the student-teacher distributions, results in stable training and good downstream performance across a number of datasets including ImageNet.

We would also like to note that the novelty in our paper extends beyond the algorithm we provide. Readers should also assess novelty in the design of the experiments we ran: to the best of our knowledge, we are the first to provide a set of experiments that show testing downstream performances with varying sizes of both pretext and labeled data and also few-shot learning experiments in the tabular datasets.



**Impact of approach**

We would like to emphasize that our method outperforms many existing methods across a wide range of experiments. We study how self-supervised methods perform when we vary the amount of pre-training data and the amount of downstream data. These experiments provide insights into the strengths and weaknesses of different self-supervised methods at different amounts of data. Across many different datasets and amounts of pre-training and downstream data, Q-Match performs the best.

Since the original submission of the paper, we have also implemented Q-Match on the ImageNet dataset. Our method outperformed both NNCLR and SWaV with **no** hyperparameter tuning on ImageNet. We developed the method and tuned the hyperparameters on tabular data in which the number of downstream classes was small ($\leq$10). But we show that Q-Match also generalizes to a different domain (image) with a large number of downstream classes (1000). We hope this convinces the reviewers that there is empirical evidence that our approach can be applied generally to different datasets and domains and is especially useful when the amount of downstream labels is limited.


**Baselines**

We appreciate the concern from the reviewers (8ARq, gtA5) that we must also compare against self-supervised techniques developed in the image domain. Although we initially focused our comparison on methods popular for tabular data (TabNet, VIME, i-Mix), we implemented many of the techniques from the image domain (SimCLR, DINO, SimSiam, and VICReg), and added these results to our paper.  We thank the reviewers for this feedback and believe these new results have strengthened our paper. It will be the first work to study the effectiveness of methods developed specifically for tabular data (TabNet, VIME, i-Mix) along with methods developed in the computer vision domain (DINO, VICReg, SimSiam, and SimCLR). Additionally we will release code which will be useful for the community to compare these self-supervised methods on different tabular datasets.

---

### Decision · Program_Chairs · 2023-01-20

**Decision:**

Reject

**Justification For Why Not Higher Score:**

Experimental results doesn't show consistent improvements over baseline. Code is not provided.

**Justification For Why Not Lower Score:**

The idea is simple and promising, and not just in tabular domain. Paper is well written.

**Metareview: Summary, Strengths And Weaknesses:**

This paper proposes using FixMatch-like distribution matching technique but for fully-unsupervised (self-supervised) learning of Tabular data. The basic model is similar to previously known non-contrastive architecture with Student and Teacher (exponential moving average of student weights). However the loss is modified to be like FixMatch. The loss matches softmax distribution of the projected student and teacher embeddings on Q directions in the embedding space. These Q directions are cached teacher embeddings from previous batches stored in a FIFO queue. The queue idea is slightly novel and simple and reviewers identified is not specific to Tabular data. The experimental results are promising but a bit underwhelming since most of the performance numbers are close to next-best baseline's numbers when incorporating the standard errors. Authors also did a series of interesting ablation studies and did additional experiments as requested by reviewers. It would be good if authors can try to extensively replicate these comparisons in non-tabular domains and provide some explanation for why this method works. Unfortunately code was not provided for reviewers to check reproducibility.